# *Mycobacterium avium* subsp. *paratuberculosis* in Sheep and Goats in Austria: Seroprevalence, Risk Factors and Detection from Boot Swab Samples

**DOI:** 10.3390/ani13091517

**Published:** 2023-04-30

**Authors:** Juliane Schrott, Eva Sodoma, Michael Dünser, Alexander Tichy, Johannes Lorenz Khol

**Affiliations:** 1Austrian Agency for Health and Food Safety (AGES), Institute for Veterinary Disease Control Innsbruck, Technikerstraße 70, 6020 Innsbruck, Austria; juliane.schrott@ages.at (J.S.); eva.sodoma@ages.at (E.S.); michael.duenser@ages.at (M.D.); 2Austrian Agency for Health and Food Safety (AGES), Institute for Veterinary Disease Control Linz, Wieningerstraße 8, 4020 Linz, Austria; 3Platform for Bioinformatics and Biostatistics, Department of Biomedical Sciences, University of Veterinary Medicine Vienna, Veterinärplatz 1, 1210 Wien, Austria; alexander.tichy@vetmeduni.ac.at; 4University Clinic for Ruminants, Department for Farm Animals and Veterinary Public Health, University of Veterinary Medicine Vienna, Veterinärplatz 1, 1210 Wien, Austria

**Keywords:** paratuberculosis, goat, sheep, risk factors, seroprevalence, boot swab samples

## Abstract

**Simple Summary:**

This study aimed to investigate the occurrence of paratuberculosis (Johne’s disease), caused by *Mycobacterium avium* subsp. *paratuberculosis* (MAP), in Austrian sheep and goats by testing 22,019 blood samples. Furthermore, detailed investigations in five MAP-infected goat herds were carried out to detect the disease in blood and faecal samples. The detected animal MAP seroprevalence was 2.0% for goats and 0.7% for sheep (calculated true prevalence 3.5% and 1.2%, respectively). Herd-level apparent MAP seroprevalence was 11.1% for goat herds and 8.9% for sheep flocks. Herds with a more intensive production system had a significantly higher risk of being infected, as well as farms with frequent trading of animals or where other ruminant species were kept on the same premise. In the five goat farms investigated, 21.8% (11.7%–28.0%, calculated true seroprevalence 38.6%) of the animals were found to be infected and 12.3% (5.0%–24.7%) of the animals were shedding the bacterium with their faeces. It was further possible to identify the bacterium using boot swab samples from the stable environment in each of the five herds. The results indicated a moderate paratuberculosis infection rate in small ruminants in Austria.

**Abstract:**

This study aimed to investigate the prevalence of *Mycobacterium avium* subsp. *paratuberculosis* (MAP) in small ruminants in Austria by testing 22,019 serum samples with ELISA for the presence of specific antibodies. Furthermore, detailed investigations in five MAP-infected goat herds were carried out by ELISA, qPCR and bacterial culture. The found animal-level apparent MAP seroprevalence was 2.0% for goats and 0.7% for sheep (calculated true prevalence 3.5% and 1.2%, respectively). Herd-level apparent MAP seroprevalence was 11.1% for goat herds and 8.9% for sheep flocks. Significant risk factors for seropositivity in goat herds were: herd size, animal trading, farmed as a dairy herd, Animal Health Service membership and cohabitation with farmed game. For sheep flocks, seroprevalence was significantly higher in flocks with animal trading and where cattle or goats were kept in the flock, respectively. The overall apparent within-herd MAP seroprevalence in the five goat farms investigated was 21.8% (11.7%–28.0%, calculated true seroprevalence 38.6%) and an overall rate of MAP shedding of 12.3% was detected (5.0%–24.7%). It was possible to identify MAP by culture using boot swab samples in each herd. The results indicated a moderate MAP infection rate in small ruminants in Austria.

## 1. Introduction

Paratuberculosis (Johne’s disease, JD) is a chronic intestinal disease caused by *Mycobacterium avium* subsp. *paratuberculosis* (MAP), affecting ruminants with a global distribution [1]. Infections in sheep and goats are mainly acquired at an early age by faecal–oral transmission of MAP, followed by a long subclinical period. The main clinical sign of JD in small ruminants is progressive weight loss [2]. Caprine and ovine paratuberculosis is known to cause considerable economic losses in affected herds [3].

In contrast to cattle, the MAP prevalence in sheep and goats is unknown in many countries and the prevalence appears to be underestimated [1]. Additionally, prevalence studies carried out among sheep and goats in Europe are scarce. Due to different study design, sampling frames and the uncertainty of the diagnostic tests, MAP prevalence among small ruminants is difficult to estimate [4]. A few studies describe several risk factors associated with MAP prevalence in small ruminants [5,6,7,8,9,10,11], but further research is needed to confirm their validity for the local small ruminant population. 

The enzyme-linked immunosorbent assay (ELISA) is a suitable cost-effective and commonly used diagnostic test for MAP prevalence studies [12]. For herd-level diagnostics and control programs in dairy herds, culture and/or polymerase chain reaction (PCR) of boot swab (BS) samples and pooled faecal samples have been applied successfully [13,14], but, to the authors’ knowledge, the application of BS sampling has not been verified for sheep and goats so far. 

In the past, cattle in Austria were predominantly affected by JD. However, MAP was also isolated from goat, sheep and wild ruminants (red deer, roe deer and mouflon) [15]. The herd-level MAP seroprevalence of Austrian cattle was observed to have increased over the years to 19.1% [16,17], but more recent studies utilising BS sampling in a region in western Austria showed a much lower MAP prevalence in dairy cattle of 0.97% [18]. In 2006, clinical JD became a notifiable disease in cattle, sheep, goat and farmed deer in Austria. Based on the compulsory national paratuberculosis control program, suspicious animals that test MAP positive have to be culled. Moreover, preventive hygienic measures have to be implemented in affected farms [19].

Sheep and goat are minor species in Austria and the production is characterised by smallholders. The Austrian national goat herd comprises 92,800 goats distributed in approximately 10,000 herds, with an average herd size of 9 animals (only 3% of the herds have more than 50 animals). About 16,000 sheep flocks hold 394,000 sheep, corresponding to an average sheep flock size of 25 animals; 13% of the flocks hold more than 50 sheep. Since the year 2011, the number of goats kept in Austria increased by about 28% and the number of sheep by about 9%, respectively [20]. According to the national animal health information system, goats are mainly kept for milk production (33%), whereas sheep are predominantly kept for meat production. The high number of animals grazing on common mountain pastures during summer months (112,000 sheep (28%) and 13,000 goats (14%) in 2020) underlines the importance of small ruminants for alpine landscape preservation in Austria [21].

The aim of the present study was to investigate the current herd-level MAP seroprevalence in Austrian sheep and goat holdings and to identify risk factors associated with a positive MAP status. Furthermore, subsequent investigations in five infected dairy goat herds were carried out to gain information about the within-herd MAP seroprevalence and bacterial shedding and whether boot swab or pooled faecal sampling may be suitable for MAP herd-level detection in goats.

## 2. Materials and Methods

### 2.1. Study Design, Animal Sampling and Data Collection 

#### 2.1.1. Serological Screening

In total, serum samples from 6434 goats in 638 herds and from 15,585 sheep in 1032 flocks were analysed in this study (Table 1). Serum sample numbers were consistent with the totality of the 2020 national annual risk-based sampling plan for the maintenance of the *Brucella melitensis*-free status in Austria [22]. In this sampling plan, herds were selected using stratified random sampling, with sample sizes proportional to the number of herds in the respective provinces (see Figure 1 and Figure 2). A sampling probability proportional to the natural logarithm of the herd size was used to select farms. In addition, dairy herds, farms with increased domestic or international livestock trade, farms using common pastures and farms with previous brucellosis history were assigned a higher selection probability. The number of samples tested for each selected herd depended on the herd size to ensure a minimum level of herd sensitivity: In herds with fewer than 12 animals, all animals were sampled. In larger herds, the sample size increased slightly. A maximum sample size of 22 animals was achieved in herds with more than 164 animals. Only sheep and goats older than 6 months were included in the sample scheme. The median herd size of the study population was 42 animals per sheep flock and 11 animals per goat herd, respectively, while the median sample size per herd was 16 samples for sheep and 8 samples for goats. Further production parameters, such as information about cohabitation with other ruminant species, common alpine pastures (yes/no), organic farming (yes/no) and Animal Health Service membership (yes/no) were obtained from the national animal health information system for the herds participating in the study (Table 2). The Animal Health Service supports its members (livestock owners) regarding animal health and welfare, providing consulting, diagnostic services, training courses and control programs. 

#### 2.1.2. Herd-Level Examination in Selected Dairy Goat Herds

In five dairy goat herds, either detected as seropositive during serological screening or during previous investigations by the National Reference Laboratory (NRL) for Paratuberculosis, detailed herd-level examinations were carried out. Herds were selected based on herd size and the willingness of the owner and the responsible veterinarian to participate in the study. Dairy goats were housed in stalls and at pasture (*n* = 4) or housed indoors only (*n* = 1). In two holdings, the herd was divided into two physically separated groups. The predominant breed was the Saanen Goat, other breeds were the German Improved Fawn, the German Improved White and the Chamois Coloured Goat. Animals did not have access to common pastures, but billy goats originating from external herds were introduced periodically to the herds. A separated kidding area was provided in three herds; moreover, kids were separated from their does after ingestion of colostrum during the first days postpartum (*n* = 4) or kids were separated immediately after birth from their does and fed with artificial colostrum (*n* = 1). In addition, two herds had dairy cattle kept on the farm.

Individual blood and faecal samples were taken from all animals older than 12 months in three herds and from a representative random sample in two herds. Blood samples were taken from the external jugular vein, according to Baumgartner and Wittek [23], using a vacuum system with serum tubes containing a clot activator (Greiner bio-one, Kremsmünster, Austria). Faecal samples were taken from the rectum; in 8 animals, no faecal sample could be collected, because faecal material was absent in the rectum at the time of sampling. Moreover, environmental BS samples and pooled faecal samples (5 animals each) were collected. BS samples were collected mainly in stable locations with high animal traffic and manure concentration in a meandering pattern, as described for cattle elsewhere [24]. Individual faecal samples were tested by qPCR, whereas BS samples and pooled faecal samples were tested by culture and qPCR, as described below. Herd size and numbers of samples per herd are shown in Table 3. For each animal, the age was documented in years, according to owners’ records, and the body condition score (BCS) was determined by a trained veterinarian, as described elsewhere [25]. Samples were transported to the laboratory for further processing immediately after collection. At the Austrian NRL for Paratuberculosis, samples were stored and refrigerated and processing started on the same working day.

### 2.2. Serological Analyses

Blood samples were first centrifuged for 5 min at 2325 g. Serum samples were tested for the presence of MAP specific antibodies, using a commercial ELISA test kit (ID Screen^®^ Paratuberculosis Indirect Screening test, ID.Vet, Grabels, France) following the manufacturer’s instructions, using the short incubation protocol. According to the data provided by the technical datasheet, the kit detects antibodies in cattle sera with a sensitivity (Se) of 56.86% and a specificity (Sp) of 99.63%; these values were applied for true prevalence (TP) calculation. Samples with positive and questionable test results were confirmed by ID Screen^®^ Paratuberculosis Indirect Confirmation test (ID.Vet, Grabels, France).

### 2.3. qPCR of Individual Faecal Samples

According to the prescribed method in the Austrian NRL for Paratuberculosis, faecal samples underwent a pre-treatment: 3 g of faeces each was put into an EZ-Drop bottle (ID Gene^®^ Easy preparation for faeces samples, ID.vet, Grabels, France), pressed carefully with a spatula and mixed for 30 s in a vortex mixer. A total of 1.5 mL of the homogenate was transferred to a 2 mL tube and centrifuged for 10 min at 13,000× *g*. Then, the supernatant was discarded and 1 mL of PowerBead Solution (Qiagen GmbH, Hilden, Germany) and 1 µL IC-DNA (bactotype^®^ MAP PCR Kit, Indical Bioscience GmbH, Leipzig, Germany) were added to the sediment. The entire suspension was transferred to a 2 mL tube with a screw cap, containing 300 mg glass beads, size 0.1 mm (Retsch GmbH, Haan, Germany), and homogenised twice for 30 s at 6800 rpm on a Precellys 24 (Bertin Technologies, Montigny Le Bretonneux, France). Then, the tube was centrifuged for 1 min at 20,000× *g* and 200 µL of the supernatant was used for the extraction. DNA was extracted using the IndiMag^®^ Pathogen Kit (Indical Bioscience GmbH, Leipzig, Germany) and the KingFisher™ Flex Purification System (Thermo Fisher Scientific, Waltham, MA, USA) or IDEAL™ 32 Extraction Robot (ID.vet, Grabels, France), depending on the number of samples, following the manufacturer’s instructions. Extracts were subjected to a commercial qPCR protocol targeting the IS900 DNA fragment of MAP (bactotype^®^ MAP PCR Kit, Indical Bioscience GmbH, Leipzig, Germany) using the following detection systems: CFX96 Touch Real-Time PCR Detection System and CFX Maestro V1-1 Software (Bio-Rad Laboratories, California, USA), Applied Biosystems 7500 Fast Real-Time PCR System and 7500 v2.3 Software (Thermo Fisher Scientific, Waltham, MA, USA), Quantstudio 6 Pro Real-Time Systems and Software 1.3.1 (Thermo Fisher Scientific, Waltham, MA, USA).

### 2.4. Culture and qPCR of Boot Swab and Pooled Faecal Samples

For analysis of the BS, a pre-treatment was applied: BS samples were homogenised for 1 min using a Laboratory Blender Stomacher 400 (Seward LTD, Worthing, UK) together with 50 mL bi-distilled water. The dissolved faeces were transferred to a 50 mL tube, centrifuged for 15 min at 2938× *g* and the supernatant was discarded. Thereafter, 3 g of pooled faecal sample or the BS manure sediment was mixed together with 30 mL 0.75% HCP solution for 30 min at 150 rpm on a laboratory shaker. Subsequently, gross particles were sedimented for 5 min. A total of 15 mL of the supernatant was transferred to another tube and incubated in the dark for 48 h at room temperature. Then, after centrifugation for 15 min at 2938× *g*, the supernatant was discarded and the sediment was re-suspended with 1 mL HPC solution. Finally, four HEYM tubes (Becton-Dickinson, Heidelberg, Germany), one M7H10 agar slope (Sigma-Aldrich, Vienna, Austria) and one liquid culture medium M7H9C, manufactured in-house as described elsewhere (Australian and New Zealand Standard Diagnostic Procedure, July 2015), respectively, were inoculated with 200 μL of each sample and incubated at 37 °C. Tubes were incubated for 12 weeks and monitored periodically for bacterial growth and fungal contamination. After four weeks of incubation, one HEYM tube of each sample was rinsed with 400 μL bi-distilled water and the liquid obtained was examined by MAP qPCR. DNA was extracted using the IndiMag^®^ Pathogen Kit and the KingFisher™ Flex Purification System and qPCR was carried out using the bactotype^®^ MAP PCR Kit, as described above. After 12 weeks, the remaining HEYM and M7H10 tubes underwent the visual evaluation. Colonies suspected to be MAP were confirmed by PCR, as described elsewhere [26]. The liquid culture samples were evaluated after 12 weeks by MAP qPCR. 

### 2.5. Statistical Analyses

Statistical data analysis was performed using the IBM SPSS Statistics v27 (IBM, New York, NY, USA) software. Potential risk factors associated with herd-level MAP seropositivity were estimated using chi-square test and logistic regression analyses and are listed in Table 2. The level of significance was set at *p* < 0.05. Operating chi-square test, the age and BCS parameters were dichotomized: BCS regular to poor were grades 1–2; BCS good to optimal were grades 3–5. Ages were also merged into two groups (age 1–4 years and 5 years or older). For the analysis of serological screening data, 10 sheep flocks with doubtful ELISA results were excluded from the dataset.

## 3. Results

### 3.1. Serological Screening

The investigation revealed an animal-level apparent MAP seroprevalence of 2.0% for goats and 0.7% for sheep (TP 3.5% and 1.2%), respectively. In 71 goat herds and 92 sheep flocks, at least one seropositive animal was detected, leading to a herd-level apparent MAP seroprevalence of 11.1% for goat herds and 8.9% for sheep flocks. In 24 goat herds, two or more samples tested positive (34%), whereas in sheep flocks 13 (14%) flocks had more than one positive sample (Table 1).

From the serological screening (ID Screen^®^ Paratuberculosis Indirect Screening test), 33 sheep and 5 goat samples showed a doubtful result. Out of these samples, 7 samples originating from sheep were confirmed positive, 11 samples remained doubtful (10 sheep and 1 goat sample) and 20 tested negative (16 sheep and 4 goat samples) by the second confirmatory testing (by ID Screen^®^ Paratuberculosis Indirect-Confirmation test). All samples testing positive at the serological screening were confirmed by the second test method.

**Table 1 animals-13-01517-t001:** Results of the serological screening for *Mycobacterium avium* subsp. *paratuberculosis* using ID Screen^®^ Paratuberculosis Indirect Screening and Confirmation test (ID.Vet, Grabels, France).

	Animals (*n*)	Seropositive Animals (*n*)	Animal-Level Apparent Seroprevalence (%)	Animal-Level Calculated True Seroprevalence (%)	Herds (*n*)	Seropositive Herds (*n*)	Herd-Level Apparent Seroprevalence (%)
**Goat**	6434	126	2.0 (95% CI = 1.6–2.3)	3.5 (95% CI = 3.1–4.0)	638	71	11.1 (95% CI = 8.9–13.8)
**Sheep**	15,585	110	0.7 (95% CI = 0.6–0.9)	1.2 (95% CI = 1.0–1.4)	1032	92	8.9 (95% CI = 7.3–10.8)
**Total**	22,019	236	1.1 (95% CI = 0.9–1.2)	1.9 (95% CI = 1.7–2.1)	1670	163	9.8 (95% CI = 8.4–11.3)

Logistic regression showed significant association between herd size and MAP seropositivity for goat herds (*p* < 0.001), but not for sheep flocks (*p* = 0.895). For goat herds, seropositivity rate was significantly higher in herds with animal trading; in dairy herds, if herd owners were Animal Health Service members and if farmed game was kept alongside goats in the herd (Table 2). For sheep flocks, seroprevalence was significantly higher in flocks with animal trading and if cattle or goat were kept together in the flock, respectively (Table 2). Depending on the region, seroprevalence ranged from 2.9 to 20.9% for goat herds and from 3.1 to 14.6% for sheep flocks, respectively (two regions with an unrepresentative sample size, each <22 herds, were not included). The prevalence in the region with the highest prevalence in both goat and sheep herds was significantly higher when compared to the prevalence in the other regions together (*p* < 0.001 and *p* < 0.004). 

**Table 2 animals-13-01517-t002:** Description of data and chi-square test *p*-values of potential risk factors for *Mycobacterium avium* subsp. *paratuberculosis* seropositivity in goat and sheep holdings in Austria.

		Goat Herds		Sheep Flocks	
**Variable ^a^**		Apparent seroprevalence in %, absolute numbers in brackets ^b^	*p*-value	Apparent seroprevalence in %, absolute numbers in brackets	*p*-value
**Animal trading ^c^**	Yes No	16.8 (25/149) 9.4 (46/489)	0.012 *	11.8 (48/406) 7.1 (44/616)	0.011 *
**Dairy herd**	Yes No	18.6 (32/172) 8.4 (39/466)	<0.001 *	6.1 (4/66) 9.2 (88/956)	0.388
**Grazing on common pastures**	Yes No	10.5 (9/86) 11.2 (62/552)	0.833	10.6 (32/302) 8.3 (60/720)	0.249
**Grazing on alpine pastures**	Yes No	9.9 (11/111) 11.4 (60/527)	0.653	10.7 (34/319) 8.3 (58/703)	0.213
**Animal Health Service membership**	Yes No	18.9 (25/132) 9.1 (46/506)	0.001 *	8.9 (24/270) 9.0 (68/752)	0.940
**Organic farming**	Yes No	13.6 (23/169) 10.2 (48/469)	0.232	8.0 (22/274) 9.4 (70/748)	0.511
**Cohabitation with other animal species**					
**Cattle**	Yes No	8.7 (23/265) 12.9 (48/373)	0.097	11.8 (34/287) 7.9 (58/735)	0.047 *
**Goat**	Yes No	-	-	13.0 (40/308) 7.3 (52/714)	0.003 *
**Sheep**	Yes No	10.1 (28/278) 11.9 (43/360)	0.456	-	-
**Farmed game**	Yes No	33.3 (7/21) 10.4 (64/617)	0.001 *	0.0 (0/10) 9.1 (92/1012)	0.318
**South American camelids**	Yes No	20.8 (5/24) 10.7 (66/614)	0.123	10.0 (2/20) 9.0 (90/1002)	0.875

^a^ variables were chosen based on their availability in the national animal health information system. ^b^ (number of seropositive herds/total number of herds). ^c^ herd entry of one or more animals in the year 2020 for replacement or breeding purposes. * *p* < 0.05 = significant.

### 3.2. Herd-Level Examination

Detailed investigations in five infected dairy goat herds (432 animals) revealed an overall apparent within-herd MAP seroprevalence of 21.8% (ranging from 11.7 to 28.0%); the calculated overall TP was 38.6%, with an overall rate of MAP shedding of 12.3% (ranging from 5.0 to 24.7%). The results are shown for each herd in Table 3 in detail. In 6 out of the 52 faecal PCR positive animals (11.5%), no seroconversion could be determined by ELISA. 

The potential risk factors of body condition score and age were analysed in terms of seropositivity and MAP shedding (data shown in Table 4). Both factors, BCS and age, were not identified as significant risk factors in the logistic regression model nor in the chi-square test (dichotomized variables as described above). However, for animals of three years old, a high frequency of MAP seropositivity (35.5%) and shedding (31.1%) was observed.

It was possible to identify MAP by culture from BS in each of the five investigated MAP positive herds. In total, 15 out of 16 BS samples tested positive (94%) on HEYM agar. Using pooled faecal samples (five animals each), 7 out of 15 samples tested positive (47%) on HEYM agar (Table 3). A minimum of one pooled faecal sample tested positive in every herd. Because of an insufficient sample amount, M7H10 was carried out for 13 BS samples (9 tested positive, 2 were contaminated) and 12 pooled faecal samples (5 resulted positive) only; M7H9C was conducted for five boot swab samples (all five positive) and nine pooled faecal samples (four with positive results; of those, two tested negative on HEYM and M7H10). 

**Table 3 animals-13-01517-t003:** Herd-level testing for *Mycobacterium avium* subsp. *paratuberculosis* in five dairy goat herds by ELISA on serum samples, individual faecal sample qPCR and culture and qPCR of boot swab and pooled faecal samples.

Herd Number	Herd Size (*n*)	Serum Sample Size (*n*)	Faecal Sample Size (*n*)	ELISA Positive ^a^	PCR Positive ^a^	Both ELISA and PCR Positive ^a^	Boot Swab Samples (HEYM) ^d^	Pooled Faecal Samples (HEYM) ^d^
**1**	114	114	112 ^c^	20.2 (23)	10.7 (12)	8.9 (10)	3/3	1/3
**2**	126	126 ^b^	122 ^c^	24.0 (30)	10.7 (13)	10.7 (13)	4/4	1/3
**3**	57	57	57	22.8 (13)	10.5 (6)	8.8 (5)	3/3	2/3
**4**	600	75	73 ^c^	28.0 (21)	24.7 (18)	23.3 (17)	2/3	2/3
**5**	550	60	60	11.7 (7)	5.0 (3)	1.7 (1)	3/3	1/3
**Total**		432	424	21.8 (94)	12.3 (52)	10.9 (46)	15/16	7/15

*Abbreviation:* HEYM—Herrold’s Egg Yolk agar slants with Mycobactin J. ^a^ results in %, absolute numbers in brackets. ^b^ 1 serum sample resulted doubtful. ^c^ in 8 animals, no faecal sample could be collected. ^d^ positive samples/number of samples.

**Table 4 animals-13-01517-t004:** *Mycobacterium avium* subsp. *paratuberculosis* ELISA and PCR results of dairy goats in different BCS classes and age groups in %, absolute numbers in brackets.

	Number of Animals	ELISA Positive	PCR Positive
**Age, years**			
1	20.5 (88)	18.2 (16)	5.7 (5)
2	19.6 (84)	13.1 (11)	8.3 (7)
3	14.5 (62)	35.5 (22)	30.6 (19)
4	10.0 (43)	16.3 (7)	18.6 (8)
5 or older	35.4 (152)	25.0 (38)	8.6 (13)
Total	100 (429)	21.9 (94)	12.1 (52)
**BCS**			
1	0.9 (4)	75.0 (3)	75.0 (3)
2	17.4 (75)	25.3 (19)	12.0 (9)
3	57.4 (248)	20.6 (51)	11.7 (29)
4	23.8 (103)	20.4 (21)	10.7 (11)
5	0.5 (2)	0.0 (0)	0.0 (0)
Total	100 (432)	21.8 (94)	12.3 (52)

## 4. Discussion

This study gives an overview about the current epidemiological situation of MAP in the Austrian small ruminant population. The investigation revealed a herd-level apparent MAP seroprevalence of 11% for goat herds and 9% for sheep flocks, respectively. The animal-level apparent MAP seroprevalence was 2.0% for goats and 0.7% for sheep. In contrast, a study conducted in Austria in 2003 reported an animal-level apparent prevalence of 0.6% in sheep and 0.0% in goats, concluding that MAP was not widespread in the small ruminant population [27]. In that study, serum and tissue samples from 169 sheep and 80 goats were investigated for MAP by ELISA and culture. One sheep was positive by ELISA and a second sheep showed positive results for MAP in PCR and culture; all the goats tested negative. The considerable difference in MAP prevalence may be partially explicable by differences in study design (the study population in the present study is more representative) and variation in the diagnostic tests used. Nevertheless, the results of our study show, that the MAP prevalence has increased in small ruminants over the last years. In Germany, the herd-level seroprevalence was estimated to be 65% in sheep flocks and 71% in goat herds [8]. According to a global review, MAP herd-level prevalence estimation determined by laboratory testing was higher than 10% in 5 out of 11 countries for sheep and in 7 out of 12 countries for goats, respectively. More than 40% of herds being affected by MAP were reported in two countries for sheep and four countries for goats [1]. However, as the number of prevalence studies carried out among sheep and goats is low and as they differ in study design and diagnostic tests used, MAP prevalence in small ruminants is still unknown in many countries and the prevalence appears to be underestimated [4]. The results of our study indicate a moderate MAP seroprevalence and compared to prevalence reported in other countries, the Austrian MAP prevalence seems to be relatively low. 

The large serological screening sample size of the present study allows flocks to be selected randomly, considering a proportional provincial distribution. As the samples originated from the national *Brucella melitensis* sampling scheme, it was not required to take blood samples for this prevalence study, which implied a benefit for animal welfare. However, large flocks, dairy herds and herds with increased animal movement were selected more frequently. As the statistical analysis showed significant association between MAP seropositivity and the risk factors’ herd size, dairy herd and animal trading in goat herds, prevalence of caprine paratuberculosis might have been overestimated in this study. 

The serological tests used in our study are validated for small ruminant sera. However, since no particular test accuracy data for sheep and goats are available for these tests, Se/Sp indicated for cattle sera had to be applied to calculate TP. The calculated TP therefore should be interpreted cautiously. MAP diagnosis using ELISA is challenging, especially in young stock, since the immune response varies between the different stages of infection [28] and between individuals [29]. Nielsen and Toft [30] reviewed test accuracies of several ELISAs for serological MAP diagnostic and concluded that test Se/Sp values differ between species. It appears that ELISA is more accurate in the detection of infection in goats than in cattle (Se ranging from 0.63 to 0.84). In contrast, Se in sheep appears to be poor (0.16–0.44). Consequently, MAP prevalence in sheep is likely to be underestimated in this study. Furthermore, test accuracy may only be valid for the target population in which a test has been validated [31]. Therefore, more research on validation of serological tests for small ruminant species is needed. The fact that a representative number of samples was investigated per herd (the median sample size per herd was 16 samples for sheep and 8 samples for goats) enabled the identification of infected flocks despite the low test sensitivity, assuming a common within-herd MAP seroprevalence of >10% in infected flocks [1]. However, infected flocks with a very low seroprevalence might have been classified as false negative.

The present study revealed a significant association between herd size and MAP seropositivity for goat herds; the same effect was not seen in sheep flocks. This finding is in contrast to what was described previously, where flock size was found to be associated with increased MAP prevalence in sheep only [8,9]. It might be assumed that association between herd size and seropositivity in goats is related also to the fact that larger goat herds in Austria are also applying an intensive production system. For sheep, the overall moderate MAP seroprevalence in Austrian flocks and the general poor test sensitivity in sheep may circumvent the detection of a potential association. We could demonstrate significant differences in flock-level MAP prevalence between regions for both species too, as described elsewhere [5,7]. Prevalence differences between regions in Austrian goat herds may be associated with an increased percentage of herds applying a more intensive production system in some geographic areas, whereas prevalence differences between regions in sheep could probably be related to a variable regional incidence of cohabitation with cattle and/or goats. Therefore, region might be considered a confounding variable. To answer this question, further studies, with a sampling scheme specifically designed for the detection of MAP, should be performed. Goat herds were significantly more often affected by MAP if they were categorised as flocks with animal trading, as a dairy herd and if herd owners were Animal Health Service members, all factors synonymous with elements of an intensive production system. These results are consistent with current knowledge: contact with other flocks [10] and intensive production systems [5,11] were reported as risk factors associated with an increased MAP seroprevalence. Furthermore, seropositivity rate was significantly higher in the present study in goat herds with cohabitation with farmed game. To the authors’ knowledge, a similar correlation has not been described before, though Verdugo et al. [32] found high flock prevalence of 46% in deer herds. For sheep flocks, the seropositivity rate was significantly higher in flocks with animal trading and if cattle or goats were kept together on the farm, respectively. These findings enable the suggestion of a possible cross-species transmission of MAP between cattle, sheep, goat and farmed game in Austria. An important tool to reconstruct infection chains and confirm MAP cross-species transmission is the bacterial strain genotyping of MAP isolates, which should be a consideration for future studies [33]. Cross species infection is suspected to occur between ruminant species [34,35,36,37,38]. Consequently, the infection of small ruminants should also be considered as a possible risk factor in the epidemiology of MAP in cattle herds. Furthermore, there is evidence for interspecies transmission between wild and domestic ruminants [15,39,40]. In an Austrian study, four out of seven MAP genotypes detected in wild ruminants were also present in local cattle and sheep herds [15]. In another study, 15 MAP genotypes across multiple wild and domestic ruminant host species were identified, leading to the conclusion that wild ruminants can act as reservoirs in the transmission of MAP to ruminant livestock [40]. Several risk factors, such as breed, kidding area, management by batches, ventilation and lactation, found to bias MAP prevalence in other studies were not considered in this study because of absent data due to the retrospective study design [5,6,10,11]. Furthermore, widespread MAP infection has been reported in goat flocks seropositive to CAEv (caprine arthritis encephalitis virus) [5]. As control programs for CAE have been implemented in Austria for many years and most herds are free of the infection, no attempts were made to survey the CAE status in participating herds in the course of our study. Both factors of age and BCS were not significant in terms of seropositivity and MAP shedding in goats in the present study, although weight loss was the main clinical sign of paratuberculosis. However, the number of investigated herds and individual animals was not sufficient to gain consolidated findings. 

In the present study, it was possible to isolate MAP by culture using BS samples in all five investigated herds (15/16 samples tested positive). Using pooled faecal samples, the detection rate was comparatively low (9/15), but all herds were detected as MAP positive by a minimum of one pooled faecal sample. According to Eamens et al. [41], pooled faecal culture is suitable for herd diagnosis if the animals are moderate or high shedders of MAP but not in herds with a few low shedders only. As two samples were positive on M7H9C, but not on HEYM in this study, it should also be considered for future investigations to use both media types for caprine samples, as goats can be infected by both MAP-S strains (sheep-type) and MAP-C strains (cattle-type) [42]. Environmental sampling is mainly used in cattle holdings as an effective and cost-efficient surveillance method [43]. Combined culture and PCR of BS samples has been applied successfully for large-scale herd-level testing in a few control programs for dairy cattle herds [13,44]. The results of the present study indicate that BS sampling may be suitable to detect MAP infection in goat herds. However, further studies are needed to validate the sensitivity of this diagnostic tool in goat holdings.

To prevent a future spread of MAP among herds and between species, a nationwide control programme should be established for sheep and goat holdings, including herd-level monitoring (e.g., by repeated environmental sample testing) and on-farm control and biosecurity measures. Some regions in Germany and Austria are currently applying such an approach in cattle [18]. Moreover, the purchase or movement of livestock should rely on herd status certification combined with an individual animal test result in order to reduce the risk of between-herd transmission of MAP by infected livestock. Goat production in Austria is a growing sector (since the year 2011 the number of goats kept has increased by about 28%), with the trend for larger flocks and more intensive production systems. Furthermore, according to experimental infection, goats have been found to be more susceptible to paratuberculosis than cattle and sheep [45]. As the prevalence was quite high in Austrian dairy goat herds, it could be judicious to focus on caprine paratuberculosis control in this production group. Windsor [2] mentioned three main approaches to control ovine and caprine paratuberculosis: management changes to decrease transmission, test and cull strategies and vaccination. Few countries are also using vaccination for disease control [1] and a meta-analysis showed vaccination reduces MAP shedding, production losses and pathology [46]. On the other hand, MAP vaccination can interfere with testing for bovine tuberculosis (TB). Austria is officially free of bovine TB, but a wildlife reservoir for *Mycobacterium caprae* in red deer exists in some geographic areas in western Austria and serves as source of infections for domestic cattle grazing on Alpine pastures during the summer [47]. In addition, vaccination may interfere with MAP control based on test-and-cull programmes using immunological tests, as ELISA is not able to differentiate between antibodies produced by infection and by vaccination [12]. Therefore, vaccination against paratuberculosis is prohibited for all ruminant species in Austria. 

## 5. Conclusions

*Mycobacterium avium* subsp. *paratuberculosis* infection is present in Austrian sheep and goat holdings, with a herd-level apparent seroprevalence of 11.1% for goat herds and 8.9% for sheep flocks. Identified risk factors for MAP seropositivity in goat herds were herd size, animal trading, dairy farms, Animal Health Service membership and cohabitation of farmed game; for sheep flocks, identified risk factors were animal trading and cohabitation with cattle or goat, respectively. Furthermore, prevalence differences were observed between different regions for both species. At the herd level, average apparent seroprevalence was 21.8% (calculated true prevalence 38.6%) and MAP shedding was 12.3% in dairy goats. It was possible to identify MAP by culture from boot swab samples in each of five MAP infected herds. 

It can be concluded, that the seroprevalence of MAP in the Austrian small ruminant population is moderate and below reported prevalence in other countries. Goat herds with an intensive production system were identified to have an increased risk to be affected by MAP; moreover, interspecies transmission seemed to occur. The use of BS sampling to identify MAP positive goat flocks may be promising but needs further investigation. Based on these results, a close monitoring of the MAP prevalence in Austrian small ruminants as well as a national control program, especially for dairy goats, seems to be advisable.

## Figures and Tables

**Figure 1 animals-13-01517-f001:**
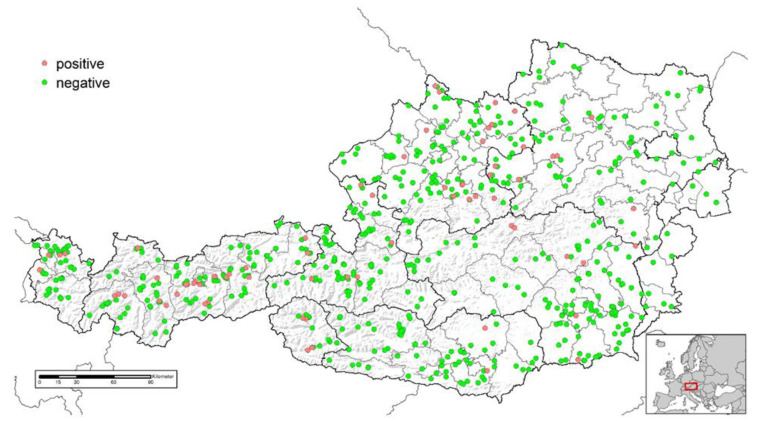
Location of goat herds investigated for the presence of *Mycobacterium avium* subsp. *paratuberculosis* specific antibodies in Austria and test results (positive, *n* = 71; negative, *n* = 567).

**Figure 2 animals-13-01517-f002:**
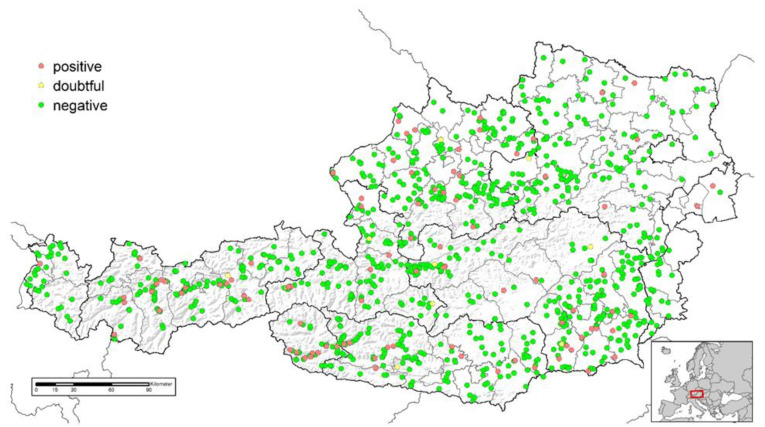
Location of sheep flocks investigated for the presence of *Mycobacterium avium* subsp. *paratuberculosis* specific antibodies in Austria and test results (positive, *n* = 92; doubtful, *n* = 10; negative, *n =* 930).

## Data Availability

The data presented in this study are available in the article.

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
