# Peer review of "Mycobacterium avium subsp. paratuberculosis in Sheep and Goats in Austria: Seroprevalence, Risk Factors and Detection from Boot Swab Samples"

_animals, 2023, doi:10.3390/ani13091517_

Round 1

Reviewer 1 Report

This manuscript investigates the prevalence of Mycobacterium avium subsp. paratuberculosis (MAP) in small ruminants in Austria by testing serum samples for the presence of antibodies by ELISA. Furthermore, specific investigations in five MAP infected goat herds were carried out by ELISA, culture and qPCR of boot swab and pooled fecal samples. 

On the other hand, a risk factors study with herd level MAP seropositivity was estimated using chi-square test and logistic regression analyses.  The identified risk factors in goat herds were herd size, animal trading, dairy farmed, AHS membership and farmed game. And for sheep flocks were as follows: animal trading and cohabitation with cattle or goat kept in the flock. 

The results indicate a moderate MAP infection rate in small ruminants in Austria. Therefore, a close monitoring of the MAP prevalence as well as a National Control Program especially for dairy goats, seems to be advisable.

In according with the authors, the use of the boot swab samples to identify MAP positive goat flocks may be promising but needs further investigation. 

The manuscript is well written and scientifically correct. Objectives have been clearly defined and Material and Methods are considered appropriate in this type of researches. Statistical analyses are necessary and well designed. Figures and tables are relevant with some recommendation. Conclusions are relevant. 

I consider this manuscript contains relevant new information about Mycobacterium avium subsp. paratuberculosis (MAP) infection in small ruminants in Austria (Europe). 

The paper could be accepted after minor changes and suggestions:

It is very important emphasize that the samples were obtained from the National Brucella melitensis Sampling Programme in Austria, which implies a benefit for animal welfare.

Some considerations: 

1. The paper should be reviewed conscientiously to correct some failures in the text (some examples are shown): 

Page 1 line 16: in in // Page 11 line 415: counties instead countries

2. Table 2 (Variables):

Could you explain what was the criterion to select the different variables?

Significant differences in flock-level MAP prevalence between regions for both small ruminants' species were detected. ¿Why was this variable not previously included?

In Austrian goat herds these significant differences may be associated to an increased percentage of herds applying a more 'intensive production' system in some geographic areas, whereas prevalence differences between regions in sheep could probably be related to a variable regional incidence of 'cohabitation with cattle and/or goats'. ¿Could 'geographic location' or 'regions' be considered a confusing variable?

The variable 'Other animals kept at the farm' could be replaced by 'cohabitation with other animal species' (cohabitation is the key word as variable). 

The variable 'animal trading' means 'animal replacement' in animal production?

What is your epidemiological explanation to relate 'AHS membership' variable as risk factor with seropositivity to MAP?

In according to reference cited below (Barrero-Domínguez et al., 2019), Why didn't you consider other variables as geographic location, breed, kidding area, management by batches, ventilation and/or type of lactation in your research?

The results of B-D et al (2019) highlight widespread PTB infection in goats flocks in Southern Spain, with intensive production system, lack of management by batches, inappropriate ventilation and seropositive to CAEv as risk factors for PTB seropositivity. In this sense, could you explain briefly what is the situation of CAEv (Caprine Arthritis Encephalitis virus) in Austria? 

Barrero-Dominguez et al (2019). Paratuberculosis in dairy goat flocks from Southern Spain: risk factors associated with seroprevalence. Vet. Rec. doi:10.1136/ vetrec-2019-105465.

3. Tables 4 and 5 must be corrected because they are unbalanced (formatted).

My suggestion is to unify both tables 4 and 5 in a single table (BCS and age risk factors)

4. References number and format is correct in according to 'Articles' from Animals Journal. Furthermore, the references chapter is well updated with two exceptions (2, 25). They should be replaced or deleted. 

Author Response

Thank you very much for the detailed, profound and constructive review of our manuscript!

Page 1 line 16: in in // Page 11 line 415: counties instead countries

A: Done

  1. Table 2 (Variables):

Could you explain what was the criterion to select the different variables?

A: These were the variables available in the national animal health information system for the herds participating in the study (line 11-112). An explanation was added to table 2.

Significant differences in flock-level MAP prevalence between regions for both small ruminants' species were detected. ¿Why was this variable not previously included?

A: We decided not to include this variable in table 2 as this would have extended the table and made it unclear. The main regional results are described in line 244-248 and in figure 1 and 2. We believe that more details about regional distribution would be of interest for Austrian readers only, but are happy to add them if requested.

In Austrian goat herds these significant differences may be associated to an increased percentage of herds applying a more 'intensive production' system in some geographic areas, whereas prevalence differences between regions in sheep could probably be related to a variable regional incidence of 'cohabitation with cattle and/or goats'. ¿Could 'geographic location' or 'regions' be considered a confusing variable?

A: A comment concerning region/location as possible confounding variable was added 363-367

The variable 'Other animals kept at the farm' could be replaced by 'cohabitation with other animal species' (cohabitation is the key word as variable). 

A: The name of the variable was changed in table 2 and where appropriate in the manuscript.

The variable 'animal trading' means 'animal replacement' in animal production?

A: The variable “animal trading” includes all kind of domestic or international livestock trade in the herd such as animal replacement and purchase of animals for breeding purposes (line 100). Information was added to table 2.

What is your epidemiological explanation to relate 'AHS membership' variable as risk factor with seropositivity to MAP?

A: AHS membership indicates a more intensive and professional production system (line 366-367).

In according to reference cited below (Barrero-Domínguez et al., 2019), Why didn't you consider other variables as geographic location, breed, kidding area, management by batches, ventilation and/or type of lactation in your research?

A: Geographic location was considered as regional differences in seroprevalence were observed and described to some extent. Unfortunately, the other variables listed were not available for this study. Line 382-384 were rephrased to emphasize this.

The results of B-D et al (2019) highlight widespread PTB infection in goats flocks in Southern Spain, with intensive production system, lack of management by batches, inappropriate ventilation and seropositive to CAEv as risk factors for PTB seropositivity. In this sense, could you explain briefly what is the situation of CAEv (Caprine Arthritis Encephalitis virus) in Austria? 

Barrero-Dominguez et al (2019). Paratuberculosis in dairy goat flocks from Southern Spain: risk factors associated with seroprevalence. Vet. Rec. doi:10.1136/ vetrec-2019-105465.

A: Done, line 385-389

  1. Tables 4 and 5 must be corrected because they are unbalanced (formatted).

My suggestion is to unify both tables 4 and 5 in a single table (BCS and age risk factors)

A: Done

  1. References number and format is correct in according to 'Articles' from Animals Journal. Furthermore, the references chapter is well updated with two exceptions (2, 25). They should be replaced or deleted. 

A: Reference 2 was deleted, 25 replaced

Reviewer 2 Report

This manuscript describes a national survey of small ruminants in Austria for MpTb infection. This could be a benchmark for future studies, national decision-making etc.

A lot of the calculation hinge on the quoted Se/Sp of the ELISA (which seem quoted as fairly high (even for cattle, unless they were clinical cases) - there is a document/poster on the ID-VET website stating Se/Sp for this ELISA in sheep (but very small numbers). Is there no citeable peer-reviewed literature?

These Se/SP values are used to calculate TP which is mentioned in Table 1 - values should be given there with 95% CI, but you only state it for animal level prevalence. Would it not equally affect herd prevalence?

And since you have gone to the trouble of calculating TP, why go back to AP in the test of the text? would it for instance change the calculations in Table 2?

Table 5 looks a bit messy on my pdf - the row with "no data"

Line 302 - refers to the previous study [27] - so that should be "that" rather than "this". Line 307 is an odd statement - don't you trust your results?

Line 340 - this is a critical statement at the heart of the study - should you have validated the test you use?

Lines 378/379 - either these are worth mentioning or delete this sentence.

Line 400 - you keep repeating this - yet ~10% of herds/flocks are affected?

Line 427 - that's back to quoting the herd/flock prevalence

There are a number of minor issues with the language, might need a further read by a native English speaker before re-submission.

I note a few for attention, there might be more:

Line 16 - double-up of "in"

Line 18 - "found" - maybe "detected"

Line 19 - risk "for" - suggest that should be "of"

Line 20 - "affected by that infection" - just condense to "infected"

Line 67 - "up" to 19.1% - that should be deleted. Presumably the prevalence is now 19.1%, not 19.1% up from a baseline of xxxx%

Line 73 - "small structured" - you use that a number of times - is that a translation from German? do you mean "smallholders"?

Line 152 - why is it "relative" - what was the comparison? usually then it would be another serological test? not a reference standard?

Line 312 - last word should be "were", not "was".

Line 319 - why would prevalence be related to them being minor species or being run by smallholders - delete

Line 350 - "oppositional" - "contrary" or "in contrast to"

Line 432 - not "On" - but "At"

Line 437 - should not be "but" - rather "and"

Author Response

Thank you very much for the detailed, profound and constructive review of our manuscript!

A lot of the calculation hinge on the quoted Se/Sp of the ELISA (which seem quoted as fairly high (even for cattle, unless they were clinical cases) - there is a document/poster on the ID-VET website stating Se/Sp for this ELISA in sheep (but very small numbers). Is there no citeable peer-reviewed literature?

A: To the best of our knowledge, no such data are available.

These Se/SP values are used to calculate TP which is mentioned in Table 1 - values should be given there with 95% CI, but you only state it for animal level prevalence. Would it not equally affect herd prevalence?

A: 95% CI was added to the table. The TP was not calculated for herds, because a different number of animals was tested in the different herds. As the Se/Sp of the test is given for single animals only, this would add another uncertain variable to the calculation.

And since you have gone to the trouble of calculating TP, why go back to AP in the test of the text? would it for instance change the calculations in Table 2?

A: We choose to stick to the AP as the calculation of the TP is based on an assumed Se of the ELISA which can not be proven and therefore stays a calculation with can be questioned.

Table 5 looks a bit messy on my pdf - the row with "no data"

A: Table 5 was changed according to the suggestions of the other Reviewer

Line 302 - refers to the previous study [27] - so that should be "that" rather than "this". Line 307 is an odd statement - don't you trust your results?

A: Line 302 changed, line 307 was rephrased

Line 340 - this is a critical statement at the heart of the study - should you have validated the test you use?

A: Of course, this would have been ideal, but unfortunately resources and samples for a validation study were not available.  That’s why this point is raised in the discussion and results were interpreted carefully.

Lines 378/379 - either these are worth mentioning or delete this sentence.

A: Information added, 388-393

Line 400 - you keep repeating this - yet ~10% of herds/flocks are affected?

A: Line (now 418) was deleted to avoid repetition and overemphasizing

Line 427 - that's back to quoting the herd/flock prevalence

A: Se comment above, we chose to stick to actual results rather than partially assumed/ questionable prevalence.

There are a number of minor issues with the language, might need a further read by a native English speaker before re-submission.

I note a few for attention, there might be more:

Line 16 - double-up of "in"

A: Deleted

Line 18 - "found" - maybe "detected"

A: Done

Line 19 - risk "for" - suggest that should be "of"

A: Done

Line 20 - "affected by that infection" - just condense to "infected"

A: Done

Line 67 - "up" to 19.1% - that should be deleted. Presumably the prevalence is now 19.1%, not 19.1% up from a baseline of xxxx%

A: Deleted

Line 73 - "small structured" - you use that a number of times - is that a translation from German? do you mean "smallholders"?

A: Changed to “characterized by smallholders”, deleted in line 324

Line 152 - why is it "relative" - what was the comparison? usually then it would be another serological test? not a reference standard?

A: “Relative” was deleted

Line 312 - last word should be "were", not "was".

A: Corrected

Line 319 - why would prevalence be related to them being minor species or being run by smallholders – delete

A: Deleted

Line 350 - "oppositional" - "contrary" or "in contrast to"

A: changed to “in contrast to”

Line 432 - not "On" - but "At"

A: changed

Line 437 - should not be "but" - rather "and"

A: changed